# Can Low-Level Exposure to Radiofrequency Fields Effect Cognitive Behaviour in Laboratory Animals? A Systematic Review of the Literature Related to Spatial Learning and Place Memory

**DOI:** 10.3390/ijerph16091607

**Published:** 2019-05-08

**Authors:** Zenon Sienkiewicz, Eric van Rongen

**Affiliations:** 1Public Health England, Centre for Radiation, Chemical and Environmental Hazards, Chilton, Oxfordshire OX11 0RQ, UK; 2Health Council of the Netherlands, P.O. Box 16052, 2500 BB The Hague, The Netherlands; e.van.rongen@gr.nl

**Keywords:** radiofrequency electromagnetic fields, spatial learning, memory, rodents

## Abstract

This review considers whether exposure to low-level radiofrequency (RF) fields, mostly associated with mobile phone technology, can influence cognitive behaviour of laboratory animals. Studies were nominated for inclusion using an a priori defined protocol with preselected criteria, and studies were excluded from analysis if they did not include sufficient details about the exposure, dosimetry or experimental protocol, or if they lacked a sham-exposed group. Overall, 62 studies were identified that have investigated the effects of RF fields on spatial memory and place learning and have been published since 1993. Of these, 17 studies were excluded, 20 studies reported no significant field-related effects, 21 studies reported significant impairments or deficits, and four studies reported beneficial consequences. The data do not suggest whether these outcomes are related to specific differences in exposure or testing conditions, or simply represent chance. However, some studies have suggested possible molecular mechanisms for the observed effects, but none of these has been substantiated through independent replication. Further behavioural studies could prove useful to resolve this situation, and it is suggested that these studies should use a consistent animal model with standardized exposure and testing protocols, and with detailed dosimetry provided by heterogeneous, anatomically-realistic animal models.

## 1. Introduction

Whether low-level exposure to radiofrequency (RF) fields could be a significant risk to public health has not been completely resolved, despite extensive epidemiological and experimental research, especially over the last 20 years. While it has been established that intense exposures can cause substantial heating of the whole body or parts of it, effects in the absence of heating remain highly controversial, and concerns continue to be expressed that exposures at environmental levels may increase the risk of some types of brain cancers, decrease male fertility or impair cognitive function [1]. Other illnesses and subjective complaints have also been attributed to exposure to RF fields. 

One of the more persistent features of the scientific debate surrounding the biological effects of RF fields has centred around the possibility that exposure may cause changes in cognitive behaviour of animals in the laboratory. More recently, suggestions have been made that exposure may also have a detrimental effect on animal behaviour in naturalistic settings. After reviewing the evidence available at the time, the World Health Organization (WHO) concluded that exposure at thermal levels could cause disruption in cognitive performance in animals in behavioural tasks, and these changes were consistent with responses to increases in body core temperature of about 1 °C or more [2]. The evidence for changes in task performance were far less well defined with exposures that did not cause hyperthermia, partly due to the paucity of data available at that time. However, it was also noted that pulsed fields with very high-peak-power pulses could affect ongoing behaviour, if specific energies per pulse exceeded the threshold for auditory perception. Such effects were attributed to very localized heating of the brain or cochlea leading to the perception of sound (the microwave hearing effect). 

Many additional laboratory studies have been published in recent years, particularly investigating the effects of RF signals associated with mobile phone technology, and it seems timely to examine the results of all these studies to inform the debate. This review focuses on papers investigating spatial memory function in rodents published since 1992, the approximate cut-off date for inclusion into the WHO monograph [2]. Studies were identified and included using a set of predetermined selection criteria (see below). Previous reviews that have covered some of this material include those from the independent Advisory Group on Non-Ionising Radiation [3,4], the International Commission on Non-Ionizing Radiation [5] and the Scientific Committee on Emerging and Newly Identified Health Risks [1,6].

## 2. Search Strategy and Inclusion Criteria

In order to identify relevant work, a predefined search strategy for published studies was formulated using PubMed and EMF Portal (https://www.emf-portal.org/en). These studies were then evaluated for inclusion against a list of exposure-related and dosimetric criteria.

PubMed was searched using the following search strategy:

(microwaves[MeSH Terms] OR extremely high frequency radio waves[MeSH Terms] OR radio waves[MeSH Terms] OR cellular phone[MeSH Terms] OR telephone, cellular[MeSH Terms] OR ((base station OR antenna) AND radiofrequency) OR mobile phone* OR cellular phone* OR cellular telephone* OR radiofrequenc* OR radio wave* OR radio-waves OR cellphone* OR cell phone* OR cellular *phone* OR mobile phone* OR microwave OR radiofrequency OR cell phone OR mobile phone OR umts OR gsm OR MHz OR ultra*wideband* OR wireless phone* OR millimeter*wave*) AND (animal OR rat OR mouse OR rats OR mice OR murine OR in vivo) AND (behaviour* OR behavior* OR memory) NOT (“in vitro”[Publication Type] OR “in vitro”[All Fields] OR ultrasound OR sound OR acoustic OR ablation OR imaging OR therap*) AND (“1993/01/01”[Date - Entrez] : “2017/12/31”[Date - Entrez]). 

EMF Portal was searched using the following keywords:

Topic: experimental studies; Frequency range: Radio frequency (>10 MHz); Mobile communications; Keywords: behaviour memory cognition; Time span: complete time span (selected 01-01-1993 to 31-12-2017)

The data were extracted by one author and checked by the other for each study. Inclusion criteria were formulated a priori: (1) The study had to include at least two exposure levels, one of which being sham exposure, with otherwise similar conditions. Standby mode of a mobile phone is not regarded as RF exposure; (2) the exposure levels had to be sufficiently controlled and documented, but a mobile phone in talk mode without control of the output level is not sufficiently controlled. The specific energy absorption rate (SAR) or other relevant exposure metrics, such as power density or electric field, and methods for determining the actual quantity had to be provided; (3) the exposures were not given in fixed order.

## 3. Results

In total, 62 publications were identified as studying the effects of RF fields on place learning and spatial memory in laboratory animals. Of these, 17 papers were excluded from further consideration because they did not comply with the inclusion criteria. These studies are listed in the Appendix A with the reason for their exclusion. The remaining 45 papers were selected for full evaluation.

Twenty papers reported at least one behavioural change that was detrimental to the performance of the task examined (Table 1). This number includes two papers from the same research group that reported identical results (and these have been treated as one study) and another three studies which also reported a significant increase in brain or body core temperature. Changed endpoints included, but were not limited to, an increased latency to locate the platform position in a Morris water maze task, less time spent in the platform quadrant during the probe trial, or an increase in the number of errors made while foraging for food in a radial arm maze.

Four studies reported field-dependent changes that may be considered beneficial or advantageous to the animal (Table 2). Of these, two papers from the same laboratory both reported that spontaneous alteration in a Y maze was increased following exposure of transgenic and non-transgenic mice. Another study reported that exposure reversed cognitive deficits in a transgenic mouse model of Alzheimer’s disease. The final paper found that exposure improved learning of the position of the escape platform in a water maze by young rats, and also improved their memory for the position of the escape platform during the probe trial.

Finally, 20 papers reported that exposure no had significant effects on place learning and spatial memory. These studies used mazes and arenas to assess behaviour in either rats or mice of various ages, and one study used the European robin to investigate magnetic orientation responses (Table 3). A few of these studies, however, reported sporadic changes in other behaviours, but no consistent changes were seen. 

In order to examine whether changes in behaviour are more likely to occur under some experimental conditions than others, the effect outcomes for the studies reported in the tables have been plotted as a function of intensity of exposure (SAR) against exposure frequency (Figure 1). The outcomes have again been grouped into three categories irrespective of behavioural task or animal species: negative effects, which had a detrimental impact on some aspect of task performance; positive effects, which resulted in a beneficial change on behaviour; and studies reporting an absence of effects, where any changes in behaviour or task performance were not statistically significant. No obvious patterns emerge from this analysis, except exposure at 2856 MHz seems to have more negative effects at higher SAR levels (above roughly 3 W/kg) and studies reporting no effects are not limited to the lowest SARs, but no account is made here for exposure duration or pattern. Figure 1 also provides an insight into the extent to which certain frequencies have received attention, and the range of SARs that have been used.

### 3.1. Narrative Review of Studies

In order to consider the outcomes of the studies from another perspective, the studies described in this narrative are not presented by outcome or by RF frequency used, but are described according to the specific behavioural task employed (i.e., the radial arm maze, Morris water maze, simple maze), and the species used (rat, mouse). The narrative has been arranged by increasing intensity of exposure (preferably using whole-body average SAR) to highlight any thresholds for effects. In this narrative, studies exposing animals during gestation or early postnatal life are considered separately from those exposing juvenile or adult animals because of the possibility that the developing nervous system or other organs are more sensitive to insult by RF fields than more mature forms. In addition, studies using transgenic animal models of Alzheimer’s disease to explore possible therapies, and a study which investigated effects on magnetic orientation in a migratory bird are also considered on their own.

#### 3.1.1. Radial Arm Maze Studies: Exposure of Juvenile or Adult Rats

Using a circular waveguide system, Lu et al. [14] exposed rats to pulsed 2450 MHz fields for 3 h per day for 30 days at a whole-body average SAR of 0.2 W/kg, and then tested their performance in a radial arm maze over 10 days. Exposure increased the number of errors made in the maze with significant differences seen on 4 of the last 6 days of training (*p* < 0.05). However, these deficits were significantly attenuated by pre-treatment with glucose. 

Lai et al. [8] exposed rats to a pulsed 2450 MHz field (2 μs at 500 pps) for 45 min per day on 10 consecutive days at a whole-body SAR of 0.6 W/kg. The SAR measured at eight locations in the brain ranged from 0.5 to 2.0 W/kg. It was found that exposed animals consistently made more errors in the radial arm maze than sham-exposed controls (*p* < 0.005). The study has been criticized [42] because differences in performance were evident between the groups on the first day of testing, suggesting possible differences in anxiety or motivation. However, it is possible that there was a very early response since the tests were performed after exposure. When the animals were pre-treated with the cholinergic agonist physostigmine or the opioid antagonist naltrexone, no field-dependent differences in behaviour were reported. Pre-treatment with another opioid antagonist, naloxone, resulted in similar differences between groups (*p* < 0.005).

Although exposure had no effect on colonic temperature of the animals, it is possible that the animals could have perceived the field, via the microwave hearing phenomenon. It has been shown that for the waveguide used, the threshold for auditory responses in the rat corresponds to an energy density per pulse of 1.5–3 µJ/cm^2^ for pulses <30 µs, corresponding to a peak power density of 0.75–1.5 W/cm^2^ [7]. The peak power density here would have been 1 W/cm^2^ suggesting that a hearing effect was possible.

Two independent groups (Cobb et al. [41], Cassel et al. [42]) failed to replicate the results of Lai et al. [8]. Both of these groups used similar numbers of animals and experimental procedures to those used by Lai and colleagues, including having restricted access to distal spatial cues normally used to perform the task. Cobb et al. [41] also pre-treated the animals with physostigmine, naltrexone or naloxone. No field-dependent effects of exposure were observed in either study.

Lai [57] proposed that residual methodological differences might support the differences in outcomes between studies. Differences included the number of choices the animals could make in the maze, with Lai limiting the number of choices to 12, whereas Cobb allowed unlimited choices (both within a 10 min trial duration). Having an increased number of choices would allow for increased learning. However, the data does not indicate that the animals used by Cobb showed over-learning, and so were unlikely to have been more resistant to any field-induced interference in learning. Additionally, the rates at which the animals in both studies reduced errors in the task were very similar, suggesting equivalent rates of learning.

Cassel and colleagues reported that exposure of rats at either 0.6 [42] or 2 W/kg [43] had no significant effect on performance in a radial arm maze. The maze used in these studies had small, transparent side walls, and so provided access to distal visual cues, but using a maze with high opaque walls (as used by Lai et al. [8]) did not affect the result [44]. It was suggested that increases in stress or anxiety in the exposed animals may have contributed to the behavioural changes originally reported by Lai et al. [8]. However, it was found that exposure had no significant effects on anxiety as measured in an elevated plus maze [45]. Animals were tested using low or high ambient light to reveal anxiogenic or anxiolytic responses respectively.

Dubreuil et al. [38] investigated the effects of head-only exposure to Global System for Mobile communication (GSM)-type pulsed 900 MHz fields. The heads of animals were exposed for 45 min immediately before behavioural testing. The animals were either foraging for food in a radial arm maze (over 10 days) or performed a food-rewarded navigation task in an open field arena, equivalent to a dry-land version of the Morris water maze (over 14 days). Different groups of animals were used for the two tasks. No significant effects on either task were seen using average SARs in the brain of either 1 or 3.5 W/kg.

Dubreuil et al. [39] investigated whether using two more-complex versions of the radial arm maze task might reveal effects of head-only exposure. In the first version of the task (lasting 12 days) a 10 s confinement period was introduced between arm choices; while the other version (lasting 16 days) also used a 15 min delay after four correct responses had been made on the last 7 days of testing.; animals were returned to their home cages during the delay. No field-dependent effects were seen in either version of the task.

Bouji et al. [53] compared the effects of repeated exposure to RF fields on behaviour in adult and older male rats. Animals were given head-only exposure to 900 MHz at a local SAR in the brain of 6 W/kg for 45 min per day, 5 days per week for 4 weeks. Spatial memory was assessed using a 4/8 version of the standard radial arm maze task during this period. The older rats showed impairments in learning (and in other behaviours) compared to the adult rats, but no field-dependent effects were seen in either adult or older rats. Exposure also had no significant effect on open field behaviour or fear conditioning in either age group. However, a field-induced decrease in anxiety-related behaviour was observed with an elevated plus maze that was independent of age. In addition, no effect of exposure was found on IL-1β, IL-6 or GFAP levels in the brain in either age group. The authors concluded that the older brain did not show increased vulnerability to RF fields. 

Ammari et al. [46] explored the effects of long-term exposure to 900 MHz GSM signals on maze performance. The heads of the animals were exposed for 45 min per day at an average SAR in the brain of 1.5 W/kg, or for 15 min per day at an SAR of 6 W/kg, 5 days per week, for 8 or 24 weeks before testing. No significant field-dependent effects were seen with either protocol. The lack of daily handling of the animals in the cage control group was considered responsible for their poorer performance.

Klose et al. [50] reported that long-term, repeated, head-only exposure of female rats to GSM signals had no effect on spatial learning at any age. Animals from 14 days old to 19 months of age were exposed 2 h a day, 5 days a week to pulsed 900 MHz at an average SAR in the brain of 0.7, 2.5 or 10 W/kg. The behaviour of animals was examined using a Morris water maze and a radial arm maze as juveniles, adults and pre-senile. No effects were observed at any age. 

In summary, 10 studies have investigated the effects of RF fields on the performance of a radial arm maze task by juvenile or adult rats. Two studies reported field-dependent changes in behaviour, and eight studies reported no significant effects. Whole-body SARs ranged from 0.2 to 10 W/kg and frequencies used were 900 or 2450 MHz. No effects threshold can be identified.

#### 3.1.2. Radial Arm Maze Studies: Exposure of Juvenile or Adult Mice

Shahin et al. [30] exposed 12-week-old mice to a 2.45 GHz continuous wave (CW) field for 2 h per day for 15, 30 or 60 days at an average whole-body SAR of 0.0146 W/kg. Beginning 15 days before the end of each exposure period, animals were trained to perform a food-reinforced task in a radial arm maze. It was found that animals made more working and reference memory errors in the maze with increasing exposure time. Animals also spent less time in the four previously-baited arms of the maze during a probe trial, with increasingly longer exposure periods having larger effects. Additional biochemical and molecular studies identified the classical hippocampal memory formation pathway as involved in these changes.

Sienkiewicz et al. [37] used a similar protocol to that used by Lai et al. [8] with rats to investigate the effects of exposure on spatial learning in mice. Animals were exposed for 45 min a day for 10 days to a pulsed 900 MHz field at an SAR of 0.05 W/kg. No significant field-dependent differences in behaviour in a radial arm maze task were observed. Animals were tested in the maze immediately after exposure or after delays of 15 or 30 minutes. In animals tested without delay (irrespective of their exposure status) there was a slightly larger variability in the time to complete the task, possibly due to some mild stress associated with the exposure situation.

In summary, two studies have investigated the effects of RF fields on the performance of a radial arm maze task by juvenile or adult mice. One study reported field-dependent changes in behaviour, and one study reported no significant effects. Whole-body SARs were 0.0146 or 0.05 W/kg and frequencies used were 900 or 2450 MHz.

#### 3.1.3. Morris Water Maze Studies: Exposure of Juvenile or Adult Rats

Deshmukh et al. [15] exposed male rats to 900 MHz for 2 h per day, 5 days per week for 30 weeks at a whole-body SAR of 0.085 mW/kg using a GTEM cell. Spatial memory integrity was then assessed with a Morris water maze. It was found that exposure resulted in significant deficits in performance of the task with animals taking significantly longer to enter the target zone in the probe trial, and the animals spent significantly less time in that zone. These differences were attributed to field-induced increases in lipid peroxidation (evidenced as increases in serum malondialdehyde concentrations) and in protein oxidation (measured as serum carbonyl content); however, glutathionine content in blood was unaffected. 

Deshmukh et al. [24] reported that low-level exposure to three different frequencies associated with mobile communications caused comparable deficits in spatial memory in rats. Male rats were exposed to 900, 1800 or 2450 MHz fields for 2 h per day, 5 days per week for 180 days at a whole-body SAR of approximately 0.6 mW/kg. At the end of the exposure period spatial memory function of all rats was tested in a Morris water maze and the brains were removed for determination of heat-shock protein (HSP) levels and DNA damage in the hippocampus. It was reported that all exposed animals exhibited deficits in spatial memory, with deficits increasing with increasing frequency, but differences between frequencies were not tested, only between each type of exposure and sham exposure. The same pattern of response was seen with an increase in HSP levels in the exposed groups: a significant difference from the sham-controls and an increasing trend with increasing frequency, but again these differences were not tested. DNA damage, assessed with the comet assay, was increased in all exposed groups, with more damage in the 1800 and 2450 MHz groups compared to that of the 900 MHz group. Improbably, identical results for all measurements were also published by Deshmuhk and colleagues following exposure of rats for 90 days [23]. This would appear to be highly probable, if not impossible, undermining the credibility of this research. However, to be as inclusive as possible, these papers are counted as one study in the present analysis.

Tang et al. [25] exposed male rats to 900 MHz fields for 3 h per day for 14 or 28 days. The SAR in the brain was 2 W/kg and the whole-body SAR was 0.016 W/kg. Spatial learning and memory were tested using a Morris water maze at the end of exposure. No behavioural effects were seen in the group exposed for 14 days, but spatial memory was impaired in the group exposed for 28 days. Signs of ultrastructural damage in the cortex and hippocampus were also reported. These effects were attributed to activation of the mkp-1/ERK pathway as significant up-regulation of mkp-1 and P-ERK/ERK proteins were observed at 28 days. 

Lu et al. [14] exposed rats to pulsed 2450 MHz fields for 3 h per day for 30 days at a whole-body average SAR of 0.2 W/kg, and then tested their performance in a Morris water maze. Compared to sham-exposed or cage control animals, exposure increased latency to locate the hidden platform in the acquisition trials in the water maze and decreased the time spent in the target quadrant in the probe trial (*p* < 0.05 in both cases). These deficits were significantly attenuated (compared to exposed animals given saline injections) when exposed animals were given glucose injections 30 min prior to each daily trial. (This study also used a radial arm maze task, and this has been described above).

Li et al. [11] exposed rats to pulsed 2450 MHz at 0.2 W/kg for 3 h per day for 30 days either with and without treatment using the glucocorticoid receptor antagonist RU468. The SAR of the brain was reported as 0.7 W/kg, but it is difficult to consider this as accurate, since the animals could move freely. Testing in a Morris water maze started 24 h after the last exposure. The escape latencies were increased on days 4–6 in the animals exposed to the field (*p* < 0.01), while in the group also treated with RU468, the escape latencies were increased on the 6th day only (*p* < 0.01, after correcting for multiple testing). In the probe trial, impairments in memory shown by the field-only exposed animals (*p* < 0.01), were partially restored by treatment with RU468. 

Wang and Lai [9] exposed rats to pulsed 2450 MHz at 1.2 W/kg for 1 h. Animals were exposed twice a day for 3 days, and performance in the maze was tested immediately after exposure It was reported that exposed animals took longer to find the platform than the sham-exposed and cage-control animals (*p* < 0.05), and spent more time trying to climb the side walls of the maze than control animals. In the probe trial the exposed animals spent less time in the platform quadrant (*p* < 0.05). It was concluded that exposure had impaired spatial reference memory and these animals had to use less efficient learning strategies. Such a conclusion may be questionable since statistical analysis of the probe trial data by one-way analysis of variance (ANOVA) revealed no significant treatment effect, although post-hoc analysis showed a statistical difference between the treatment groups.

The effects of simultaneous exposure to a RF EMF and a temporarily incoherent magnetic field were investigated by Lai [10]. Rats were exposed for 1 h to a continuous wave 2450 MHz field at 1.2 W/kg using a cylindrical waveguide system inside a set of Helmholtz coils. These coils were used to generate a ‘magnetic noise’ that consisted of a complex low frequency magnetic signal at a flux density of 6 µT. It was found that the time taken to locate the escape platform was significantly increased after exposure to the RF field (*p* < 0.001). This increase in escape time was reduced, following simultaneous exposure to the RF field and magnetic noise, but remained significant (*p* < 0.016). There was no effect of the magnetic noise alone. Exposure to the RF field alone resulted in the animals spending significantly less time during the probe trial in the platform quadrant (*p* < 0.05).

Kumlin et al. [31] exposed groups of juvenile rats to 900 MHz GSM signals for 2 h/day, 5 days/week for 5 weeks at a whole-body SAR of 0.3 or 3 W/kg. It was found that exposure at both SAR values significantly improved the learning of the location of the escape platform (compared to sham-exposed controls) and the group exposed at the higher SAR also showed improved memory in the probe trial. These cognitive changes were not reflected in other behavioural tests and no significant effects were observed in an open field, elevated plus maze or acoustic startle response test. 

Tan et al. [27] compared the effects of single, acute exposure to two different frequencies on performance of rats in a Morris water maze task. Rats were exposed to either 1500 or 2856 MHz for 6 min or exposed to both frequencies for 6 min each at a whole-body SAR of 1.8 or 1.7 W/kg or at 3.7 or 3.3 W/kg. Escape times in the maze were measured on days 1, 2, 7, 14 and 28 after exposure. It was found that exposure increased escape latency in the task only at the higher SAR, irrespective of frequency on all days. Changes were also reported in the EEG and in the morphology of the hippocampus of animals exposed at the higher SAR. 

Wang et al. [28] trained male rats to locate a hidden platform in a Morris water maze over 3 days. Animals were then exposed to a pulsed 2856 MHz field at an average SAR in the brain of 1.7, 3.5 or 7.5 W/kg for 6 min per day, 5 days per week for 6 weeks. Additional water maze trials with a platform present were given from 6 h to 12 months after exposure finished, and a probe trial was conducted 3 days after exposure. It was found that only exposure at 7.5 W/kg caused deficits in learning and memory: the escape latencies of these animals were significantly increased 7 days after exposure and at 1, 3 and 9 months after exposure, and the animals spent less time in the target quadrant and made fewer crossings of the platform location in the probe trial. Swimming speed and body temperature of the animals were not affected by any exposure. Changes in the EEG, NMDA receptor subunits and hippocampal ultrastructure were also found following exposure at 7.5 W/kg.

Li et al. [21] investigated the effects of repeated exposure to pulsed 2856 MHz microwaves on maze behaviour. Young male rats were exposed beginning at 4 weeks of age for 6 min three times a week for 6 weeks at average power densities from 5 to 30 mW/cm^2^ (corresponding to whole-body average SARs of about 1.5 to 9 W/kg). An increased escape latency was found after exposures at 5 mW/cm^2^ at 14 days, after 10 mW/cm^2^ at 4, 14 and 28 days, and after 20 and 30 mW/cm^2^ at 3, 4, 14 and 28 days (*p* < 0.05). All exposure groups spent significantly less time in the target quadrant in a probe trial conducted 5 days after exposure (*p* < 0.05, in all cases). Exposure had no effects on swimming speed at any time.

In a series of studies from the same group, it has been reported that single exposure for only a few minutes to high-power pulsed 2856 MHz microwaves can cause lasting changes in maze behaviour and hippocampal function in male rats. Wang et al. [16] reported that exposure for 6 min at average SARs in the brain of 35 W/kg (causing a peak rise in brain temperature of 1.2 °C and of 0.6 °C in the body of anesthetized animals) significantly increased the time to locate the platform in the Morris water maze up to 24 h after exposure; exposure at 7 W/kg only had a significant effect after 6 h, and exposure at 3.5 W/kg had no effect. Exposures at 7 and 35 W/kg caused a significantly decreased memory for the location of the platform during the probe trial conducted 72 h after exposure. Qiao et al. [17] reported changes in Morris water maze behaviour at 1, 2, 3 and 7 days after 5 min pulsed 2856 MHz exposure (*p* < 0.01–0.05) at a whole-body SAR of 14 W/kg (causing a 0.3 °C rise in body temperature measured before and after exposure). Exposed animals took longer to locate the platform in acquisition trials in the maze. No differences between exposed and sham-exposed groups were observed at 4 and 14 days after exposure. Wang et al. [20] reported that similar changes in maze behaviour persisted for 18 months after the same type of exposure for 6 min at a whole-body SAR of 15 W/kg, local SAR in the brain of 35 W/kg.

In summary, 14 studies have investigated the effects of RF fields on the performance of a water maze task by juvenile or adult rats. Thirteen studies reported a field-dependent deficit in behaviour (with two of these studies suggesting effect thresholds of around 3 W/kg and another suggesting effects only with prolonged exposure durations) and one study reported an improvement in learning. Whole-body SARs ranged from 0.085 mW/kg to 35 W/kg and frequencies used were 900, 1800, 2450 or 2856 MHz.

#### 3.1.4. Morris Water Maze Studies: Early Life Exposures of Rats

Cobb et al. [35] exposed pregnant rats to ultra-wideband (UWB) pulses (55 kV/m peak, 1.8 ns pulse width, 300 ps rise time, 1000 pulses per second, 0.1–1 GHz, SAR 45 mW/kg). The exposure was 2 min per day during gestation days 3–18 and was continued during 10 postnatal days for some animals. Exposure had no effects on performance of a Morris water maze task by adult male offspring. The medial-to-lateral length of the hippocampus was longer in exposed pups (*p* = 0.001) but the authors did not consider this change to represent a field-dependent effect. Using lead acetate as a positive control caused significant effects in numerous endpoints. 

Daniels et al. [47] exposed neonatal rats for 3 h per day from postnatal day 2 to 14 to an 840 MHz field at a power density of 60 µW/m^2^. No effects of exposure were observed on memory function when the animals were tested at 58 days of age, but an increase in freezing behaviour was observed in males, which was considered indicative of mood disturbance. 

Takahashi et al. [48] exposed pregnant rats during gestation and the progeny during lactation to 2140 MHz RF EMF for 20 h per day at two exposure levels. At the higher exposure level, the average SAR was about 0.1 W/kg for the dams, and slightly higher for the foetuses and the progeny. At the lower level, the SARs were about 43% of these. Several variables were measured, including memory function of the first-generation offspring. It was found that exposure had no effect on activity in an open field arena at 5 and 8 weeks of age, or on performance in a Morris water maze at the age of 9 weeks.

Shirai et al. [52] exposed three generations of Sprague Dawley rats to a 2.14 GHz Wideband Code Division Multiple Access (W-CDMA) mobile phone signals. The animals were exposed for 20 h per day at SAR levels of not more than 0.24 W/kg (high), not more than 0.08 W/kg (low) or 0 W/kg (sham). The exposure levels varied over the duration of the study, since the animals were exposed in utero, as pups and as juveniles. Pregnant animals were exposed from gestational day 7 to weaning and then their offspring (both males and females) were continuously exposed until 6 weeks of age. At 11 weeks of age, the offspring were mated, and the procedure repeated. This was again repeated with the third generation. The whole experiment was performed in duplicate. Offspring were tested for developmental indices (at 6–12 days old), in an open field (at 7 weeks old) and in a Morris water maze (at 9 weeks old). No consistent effects of exposure were observed in the water maze or on any of the other developmental endpoints investigated in any of the three successive generations.

As part of a multigenerational teratology study, Shirai et al. [56] investigated effects on cognitive behaviour. Rats were exposed during gestation and early life to a mixture of eight different communication signals, ranging in frequency from 880 MHz to 5.18 GHz. Animals were exposed for 20 h per day from gestational day 7 until 6 weeks of age at a whole-body SAR of approximately 0.08 or 0.4 W/kg. When animals were tested in a Morris water maze at 9 weeks of age, it was found that exposure had no significant effect on acquisition, although memory for the location of the escape platform was slightly but significantly impaired in the males exposed at the higher SAR in one measure (time in target quadrant) but not in another (crossings of platform location). For this reason, this result in the males was discounted by the authors. In other tests on dams and offspring, exposure had only a few sporadic positive effects. Overall, multi-frequency RF field exposure was not considered to have had any adverse effects on pregnant animals or their offspring. 

In summary, five studies have investigated the effects of prenatal or early postnatal exposure to RF fields on the performance of a water maze task by adult rats. All studies reported no significant effects. Whole-body SARs ranged from 45 mW/kg to 0.4 W/kg (one study used a power density of 60 µW/m^2^) and frequencies ranged from 800 to 5180 MHz, plus UWB pulses of 0.1–1 GHz.

#### 3.1.5. Morris Water Maze Studies: Exposure of Juvenile or Adult Mice

Shahin et al. [26] exposed 12-week-old mice to a 2.45 GHz CW field for 2 h per day for 15, 30 or 60 days at an average whole-body SAR of 0.0146 W/kg. The rectal temperature of the animals was not significantly increased by this treatment. Groups of animals were tested in a Morris water maze after exposure, and it was found that all exposed animals showed an impairment in the acquisition of the task, taking significantly longer to locate the hidden platform in the acquisition trials, and the number of animals failing to locate the platform in the allotted time was also significantly increased. Similarly, during the probe trial, all the exposed animals showed impaired retention for the location of the platform by spending significantly less time in the target quadrant, and visiting the previous location of the platform fewer times. Further, the magnitude of these deficits was related to the exposure time and were largest in the animals exposed for 60 days and smallest in the animals exposed for 15 days. Additional trials using a visible platform suggested that these impairments could not be explained by visual or motor deficits. It was also found that exposure affected hippocampal neuronal morphology and caused increased oxidative/nitrosative stress leading to increased apoptosis. 

Chaturvedi et al. [13] exposed young adult, male mice to continuous wave 2450 MHz microwaves for 2 h per day for 30 days at a whole-body average SAR of 0.03 W/kg. A Morris water maze task was performed on days 17 to 22 of exposure. It was reported that exposure had no effect on acquisition of the task, since the time to locate the hidden platform was not different between treatment groups, although it is not clear what the data presented represent, since the escape latencies of both groups are around 90 s, yet the maximum latency was defined at 20 s. During the probe trial, the exposed animals spent significantly less time in the target quadrant (*p* < 0.05) suggesting exposure had impaired the memory of the platform location. However, modest group sizes and the use of what appears to be a non-standard testing protocol make drawing any conclusions doubtful. Behaviour in an activity wheel in the home cage was continuously monitored in these animals (except for 3 h per day to allow exposure) for 12 days prior to exposure, 7 days after exposure started and the last 7 days of exposure. During pre-exposure the phase angle difference between lights-off and the onset of activity was about +1 h for all animals. This did not substantially change after 7 days of exposure but was significantly increased to almost +3 h after 30 days of exposure (*p* < 0.05) indicating a clear shift in activity towards the light phase. In addition, exposed mice were less active during the dark phase. Overall, long-term exposure seemed to have disrupted the normal activity cycle. Sharma et al. [19] reported that exposure of mice to 10 GHz microwaves at a whole-body SAR of 0.18 W/kg for 2 h a day for 30 days significantly increased the time to locate the escape platform in a Morris water maze task (*p* < 0.001). Following two habituation trials in the maze, mice were tested twice a day for 6 days in the maze. Both exposed and sham-exposed groups reduced their mean escape times each day, but the exposed group was consistently slower by about 10 s every day. Unfortunately, no final probe trial (without the platform being present) was conducted to assess spatial memory in these animals.

Sharma et al. [29] exposed 2-week-old mice to a 10 GHz field for 2 h per day for 15 days at a whole-body SAR of 0.179 W/kg. Performance in a Morris water maze was investigated approximately 2 weeks after exposure. Significant deficits were reported in learning during acquisition trials and in memory during the probe trial, indicating exposure had a sustained effect on behaviour. Additional studies suggested exposure had also caused significant biochemical and histopathological changes in the brain, as well reducing both body and brain weights immediately after exposure. The authors suggested that these results showed the increased sensitivity of the juvenile brain to insult by RF fields. 

Zhang et al. [55] reported that exposure of juvenile mice to 1800 MHz had no effect on spatial learning or memory as assessed in a Morris water maze. Animals were exposed for 6 h each day for 28 days at a whole-body SAR of 2.7 W/kg (2.2 W/kg in the brain). Additional behavioural tests indicated that exposure increased levels of anxiety as measured in an elevated plus maze, and also increased levels of γ-aminobutyric acid (GABA) and aspartic acid (Asp) in the cortex and hippocampus. Depression-like behaviour was not affected.

In summary, five studies have investigated the effects of RF fields on the performance of a water maze task by juvenile or adult mice. Three studies reported field-dependent changes in behaviour, another reported an equivocal effect, and one study (using the highest SAR) reported no significant effects. Whole-body SARs were 0.0146 to 2.7 W/kg and frequencies used were 1800 or 2450 MHz.

#### 3.1.6. Morris Water Maze Studies: Early Life Exposures of Mice

Zhang et al. [22] exposed mice for most of pregnancy (from gestation day 3.5 to day 18) for 12 h per day to a 9.417 GHz field with an intensity of 200 V/m. The authors quote an SAR level of 2 W/kg, but do not indicate how this was determined. The offspring were assessed at 5 weeks of age for cognitive function using a Morris water maze. It was found that male mice showed a significant decrease in task acquisition and showed an impairment in the memory of the location of the escape platform during the probe trial. Female mice did not show these changes, however, suggesting gender-dependent effects on spatial memory. In contrast, other tests indicated that exposure caused higher levels of behavioural anxiety in males and females, as well decreasing depression-like behaviour in both sexes.

#### 3.1.7. T-Maze Studies: Exposure of Juvenile or Adult Rats

Yamaguchi et al. [40] investigated effects on reversal learning in a T-maze following exposure of rats to pulsed 1439 MHz Personal Digital Cellular (PDC) signals. In a 4-day experiment, the animals were exposed for 1 h per day at a whole-body SAR of 1.7 W/kg, or 45 min per day at a whole-body SAR of 5.7 W/kg immediately before testing. In a 4-week experiment, animals were exposed at 1.7 W/kg for 1 h per day for 5 days a week; testing was performed in the 4th week after each exposure. No effects were observed after exposures at the lower SAR level, and these had no effect on body temperature. However, performance was significantly impaired after exposures at the higher SAR level (*p* < 0.001), and these increased body core temperature by up to 2 °C. 

#### 3.1.8. Studies Using Juvenile or Adult Transgenic Animals

Mori and Arendash [32] reported that repeated exposure of Alzheimer’s transgenic mice or non-transgenic littermates to 918 MHz GSM signals increased spontaneous alteration in a Y maze. Animals were exposed twice a day for a month at 17–35 V/m and tested in the maze for a single, 5 min trial. Unfortunately, data were not reported by genotype, but by combining both transgenic and non-transgenic animals into a single group, it was found that there was a 26% increase in percentage alternation (*p* < 0.05) in the exposed mice compared to the sham-exposed mice. This suggests that exposure had increased levels of exploration in the mice, irrespective of their genotype. 

In a follow-up study, Arendash et al. [33] reported that in a combined group of transgenic and non-transgenic animals, similar exposures for up to 2 months had no significant effects on performance of spatial memory tasks, although alternation in a Y maze task was again shown to be increased by exposure. Exposure also had no measurable effect on activity, exploration or balance of these animals.

Jeong et al. [34] exposed mice transgenic for the expression of several Alzheimer-related proteins and for a rapid development of amyloid β plaques. Young female transgenic mice and their wildtype (WT) counterparts were exposed to 1950 MHz fields at 5 W/kg for 2 h per day, 5 days per week for 8 months. The maximum increase in rectal temperature was 0.5 °C. Decreases in alternation in a Y maze, and other behavioural changes shown by unexposed transgenic animals were significantly reversed by exposure of the transgenic animals. Histopathological analysis of these brains indicated that the development of amyloid β plaques and other parameters associated with Alzheimer’s diseases were also decreased. No such changes were observed in exposed WT mice. The authors speculated that exposure to RF fields might be beneficial against the development of Alzheimer’s disease.

Son et al. [54] exposed female 5xFAD transgenic mice to a 1950 MHz field for 2 h per day, 5 days per week for 3 months at a whole-body SAR of 5 W/kg. After the last exposure, several behavioural tests were performed, and the levels of the Alzheimer proteins in the brain and blood were assessed. No effects of exposure were observed on spatial memory functions assessed using a Y maze or a Morris water maze, and exposure was without significant effect on behaviour in an open field or on performance of a novel object recognition task.

In summary, four studies have investigated the effects of RF fields on spatial learning by transgenic animals used as a model for Alzheimer’s disease. Three studies reported field-dependent improvements in behaviour in a Y-maze, and two studies reported no effect on the performance of a radial arm maze task. Two studies used an electric field strength of 17–35 V/m and two studies used a whole-body SAR of 5 W/kg. Frequencies used were 918 or 1950 MHz.

#### 3.1.9. Navigation in a Migratory Bird

Seasonal migration across long distances is possibly the most dramatic example of place navigation in animals. It has been known for many years that some birds (and other animals) have a specialized neural apparatus that allows the use of the Earth’s magnetic field as a directional cue [58]. Engels et al. [51] reported that exposure to the background electric and magnetic fields in the urban environment affected the orientation behaviour of migratory European robins, *Eritacus rubeula*. In laboratory tests using conical arenas that were lined with scratch-sensitive paper to record movement, the birds did not orientate as expected towards magnetic north (their normal migratory direction in the spring). However, the robins were able to orientate towards north after grounded aluminium plates had been installed to shield the testing environment (*p* < 0.001). These plates reduced the magnetic fields in the frequency range of about 50 kHz to 5 MHz by two orders of magnitude (unshielded: 1.8 nT, shielded: 2.56 nT). In addition, the birds changed their orientation when the components of the static magnetic field were rotated or inverted (*p* = 0.008). In further tests, birds in the shielded condition were disorientated when exposed to specifically-generated broadband fields from 2 kHz to about 5 MHz, with a magnetic field strength of 278 nT. The disorientation response could also be produced using fields of about 20 to 450 kHz (133 nT) and about 0.6 to 3 MHz (34 nT), suggesting the response was not limited to a single or narrow frequency band.

## 4. Discussion

Overall, the publications included in the review form a highly heterogenous group, both in terms of experimental parameters and behavioural changes. The most used method in this group was the Morris water maze, which involves animals having to learn the location of a submerged escape platform in a small swimming pool: 66% of the papers examined used this maze. The radial arm maze was also much used, and this requires an animal to learn the location of food rewards placed at the ends of the arms of the maze: 27% of the papers examined used this maze. Both mazes are widely used for examining spatial learning and memory in rodents [59,60]. The first study in this group of papers used a radial arm maze, and this prompted a number of investigations using this apparatus. However, the use of this maze has substantially declined since then, and more recent studies have mostly used the Morris water maze. Additionally, a few studies have used dry versions of the water maze task using an open field or have used a Y maze. No particular maze or task has provided a consistent response or outcome in these studies, and both impairments and no effects have been reported with the water maze and radial arm maze. 

In addition, the studies reporting a significant effect of exposure used a range of frequencies that included 900, 1800 and 2450 MHz, with both continuous wave and pulsed fields; whole-body SARs ranged from as little as a 0.1 mW/kg to more than 10 W/kg, and exposure schedules were highly variable in terms of hours per day of exposure and number of days of exposure. Most studies restrained their animals to prevent movement during exposure (to ensure better dosimetric control), but animals were free to move in some studies.

The studies which reported an absence of effects also form a very heterogenous database, in terms of frequency, SAR, and exposure schedule used. Frequencies used again included 900, 1800 and 2450 MHz, with whole-body SARs ranging from less than 0.1 to over 10 W/kg. Animals were also exposed from a few minutes a day to several hours a day over several days or weeks. As with the studies reporting effects, the majority of studies that reported an absence of effects restrained their animals during exposure, but animals were freely-moving in other studies. 

Due to the heterogeneity in experimental protocols used in these studies, a comparison between the studies summarized in Table 1, Table 2 and Table 3, Figure 1 and described in the narrative does not suggest simple explanations for the differences in outcomes in terms of behaviour examined, frequency or signal used, or the amount energy absorbed by the animals. Additionally, within the limitations of the existing database, neither the sex, age or weight of the animals appear to be a crucial factor in determining the outcome. However, it is noteworthy that several groups who attempted to confirm the results of the first study to report field-induced deficits in radial arm maze performance of rats were not successful in replicating the reported effects in either rats or mice. Some of these studies aimed to use an identical protocol to that used in the original study, while others tried to improve on some aspect of that protocol. In none of these studies, however, was there any suggestion of a field-induced deficit. 

## 5. Conclusions

This review was undertaken to help to answer the question of whether exposure to RF fields was able to consistently and reliably affect cognitive behaviour of animals. A total of 45 papers published since 1993 were identified that had investigated spatial learning and memory in laboratory tests and met certain predefined quality criteria for inclusion.

However, after reviewing the evidence and comparing the studies, it is not yet possible to give an unequivocal answer to that question. Using a weight-of-evidence approach indicates that there have been the same number of studies published since 1993 that have reported an adverse effect on spatial learning and memory (20 papers, treating two papers reporting identical results as one study) as have not reported an effect (20 papers), while a far smaller number of studies (four papers) have reported an improvement in performance, mostly offsetting age-related decrements in performance. Some studies have suggested possible molecular mechanisms for the observed effects, but none of these possibilities has yet been substantiated through independent replication.

Assuming that all the studies were conducted with equal diligence and precision, and all results are reported honestly and impartially, and it is not obvious why similar studies should have sometimes reported conflicting and opposing outcomes. If these differences are not simply due to chance, it must be assumed that these outcomes are somehow related to the use of differing experimental or exposure parameters, but it is not possible to identify those parameter(s) with any degree of certainty. The ambiguity in outcome may relate to subtle differences in the genetic background or health status of the animals, how the animals were housed and maintained, or related to how the behavioural tasks were performed, but equally possible are inadequacies or other problems in some of the studies which have not yet been identified. 

Overall, this ambiguity does not provide strong support to the hypothesis that low-level RF fields can impair cognitive behaviour in laboratory animals, but neither does it provide unequivocal evidence for an absence of effects. Since a few studies reported behavioural changes that appear beneficial to the animals, it is also possible that exposure under some circumstances may improve cognitive ability and task performance. However, what these conditions might be are far less well defined due to the paucity of data. 

It is therefore suggested that additional basic research is required. To avoid exacerbating the present situation which has a plethora of models and protocols, it is suggested that this behavioural work should use a single animal model with standardized exposure and testing protocols to ensure increased consistency between studies. Without trying to be prohibitive or too restrictive, it is suggested that
Possible confounding resulting from immobilization stress caused by restraining the animals during exposure should be avoided, so exposure systems, like reverberation chambers which can produce well-controlled and repeatable SARs without the need for restraint, are recommended.A single whole-body SAR value should also be avoided, and experiments should use a range of whole-body SARs from 0 to 5 W/kg or more (depending on species) to provide some indication of the dose-response relationship. If possible, an appropriate positive control group should also be included. However, exposures causing overt heating of the animal, where body core temperatures are raised by 1 °C or more should be avoided in order to prevent any confounding from thermal effects.Of particular importance with studies using RF fields is the necessity for detailed computational dosimetry in heterogeneous, anatomically realistic animal models to determine both the average absorbed energy for the whole animal, and the spatial distribution of the absorbed energies at the organ and tissue level. Unfortunately, not all recent papers (see studies listed in Appendix A) provide sufficient information to allow a reasonable estimate of the SAR to be made.Given the ubiquity of the Morris water maze in many laboratories investigating neurobehavioral toxicology and teratology, the use of this maze for these studies seems most appropriate. The task is suitable for animals of all ages, including weaning mice [61] and is amenable to using the many commercial systems that allow automatic recording and analysis of the animals’ behaviour in the maze, so helping to eliminate a potential source of subjective error. The data in the computer files can also be readily re-analysed if necessary.

The numbers of animals used in many experiments are modest, with group sizes of around 6–12, so larger numbers of animals (of around 12–18 animals) should be used to give improved confidence in the results obtained. Samples of brain tissues from experimental animals should be taken and stored after any experiment to produce a bank of material that could be shared among research groups and examined for molecular or cellular changes.

## Figures and Tables

**Figure 1 ijerph-16-01607-f001:**
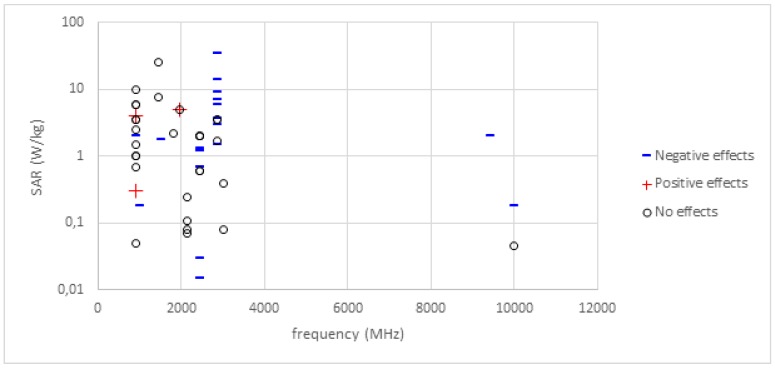
Diagram showing the exposures and RF frequencies used by the studies listed in Table 1, Table 2 and Table 3, irrespective of behavioural task or species. Exposures are expressed as whole-body specific energy absorption rate (SAR) in W/kg. Studies are shown as either having a detrimental effect on behaviour (Negative studies) a beneficial effect (Positive effect) or not having a significant effect on outcome (No effects).

**Table 1 ijerph-16-01607-t001:** Behavioural studies with radiofrequency (RF) fields reporting significant impairments of place learning and spatial memory.

Model	Exposure	Response	Comment	Reference
12-arm radial maze SD rat (*n* = 8) 250–300 g	2450 MHz, pulsed; 2 μs pulses at 500 pps 45 min/day, 10 days WBA SAR 0.6 W/kg Brain local SAR 0.5–2.0 W/kg Restrained	RF alone: more errors than sham. Pre-treatment with physostigmine or naltrexone: no difference exposed/sham. Pre-treatment with naloxone: no effect	Behaviour assessed after each daily exposure. SAR measured according to Chou et al. [7]	Lai et al. [8]
MWM SD rat (*n* = 11, 12) 2–3 months 250–300 g	2450 MHz pulsed; 2 μs pulses at 500 pps 60 min × 2/day, 3 days WBA SAR 1.2 W/kg Restrained	Increased escape times, no effect on speed; less time in correct quadrant during probe trial	Differences in probe trial not significant using ANOVA, but significant using Newman–Keuls post-hoc analysis. SAR measured according to Chou et al. [7]	Wang and Lai [9]
MWM SD rat (*n* = 8) 2–3 months 250–300 g	2450 MHz CW 60 min × 2/day, 3 days WBA SAR 1.2 W/kg Temporally incoherent magnetic noise at 6 µT Restrained	Increased escape times, less time in correct quadrant during probe trial; smaller changes after co-exposure with magnetic noise	Magnetic noise alone had no effect. SAR measured according to Chou et al. [7]	Lai [10]
MWM Wistar rat (*n* = 5) 3 months	Pulsed 2450 MHz ± glucocorticoid receptor antagonist RU468 3 h/day, 30 days Brain SAR 0.7 W/kg; WBA SAR 0.2 W/kg Free	RF: increased escape latency on day 4–6; RF + RU468: on day 6. RF: impaired memory RF + U468: no effects on memory	SAR calculated by calorimetry. Brain SAR seems doubtful	Li et al. [11]
MWM, activity wheel Parkes mouse (*n* = 5) 40 days	2450 MHz CW WBA SAR 0.03 W/kg 120 min/day, 30 days Restrained	No effect on escape latency during acquisition; less time in correct quadrant during probe trial. Phase shift in activity after 30 days, less active at night	Mice given 20 s to locate the escape platform Size of pool not reported. Modest group sizes, non-standard testing protocol. SAR calculated according to Gandhi [12]	Chaturvedi et al. [13]
MWM, 8-arm radial maze Wistar rat (*n* = 6) 3 months	2450 MHz, pulsed 10 µs, 800 pps WBA SAR 0.2 W/kg Brain SAR 0.7 W/kg 3 h/day for 30 days Free	Deficits in both tasks reduced by i.p. injection of glucose before each trial	SAR calculated using calorimetry	Lu et al. [14]
MWM Fischer rat (*n* = 6) 150–200 g	900 MHz WBA SAR 0.085 mW/kg 2 h/day, 5 days/week for 30 days Restrained	Increased time to locate, and decreased time in target quadrant in probe trial	SAR used is very low	Deshmukh et al. [15]
MWM Wistar rat (*n* = 10) 8 weeks	2856 MHz, pulsed 200 or 500 pps, pulse width 500 ns 6 min Average brain SAR 3.5, 7 or 35 W/kg Restrained	Increased escape latency at 6 h after exposure at 7 or 35 W/kg; and for 24 h after exposure at 35 W/kg. Reduced number of crossings of platform location in probe trial at 7 W/kg and 35 W/kg	Highest SAR caused a rise in brain temperature of 1.2 °C and in rectal temperature of 0.6 °C of anesthetized rats. SAR calculated using FDTD methods	Wang et al. [16]
MWM Wistar rat (*n* = 15) 200 g	2856 MHz, pulsed 500 pps pulse width 500 ns 5 min WBA SAR 14 W/kg Restrained	Increased escape latency 1 day, 2, 3 and 7 days after exposure. Non-significant increase after 6 h, 4 and 14 days.	Rise in body temperature of 0.3 °C. No probe trial. SAR calculated using FDTD methods	Qiao et al. [17]
MWM Swiss albino mouse (*n* = 6) 6–8 weeks	10 GHz WBA SAR 0.18 W/kg 2 h/day, 30 days Restrained	Increased escape latency	Two mice exposed together in same cage. No probe trial. SAR calculated according to Durney et al. [18]	Sharma et al. [19]
MWM Wistar rat (*n* = 15) 8 weeks	2856 MHz, pulsed 500 pps, pulse width 500 ns 6 min Brain SAR 35 W/kg WBA SAR 15 W/kg Restrained	Increased escape latency up to 18 months after exposure	Rise in brain temperature of 1.2 °C and in rectal temperature of 0.6 °C of anesthetized rats. No probe trial. SAR calculated using FDTD methods	Wang et al. [20]
MWM Wistar rat (*n* = 15) 4 weeks	2.856 GHz 6 min × 3 per week, 6 weeks PD 5, 10, 20 or 30 mW/cm^2^ (50, 100, 200, 300 W/m^2^) Restrained	Escape latency increased at 5 mW/cm^2^ at 14 days, at 10 mW/cm^2^ at 4, 14, 28 days and at 20 and 30 mW/cm^2^ at 3, 4, 14, 28 days after exposure. All exposed groups spent less time in the target quadrant in probe trial 5 days after exposure, and escape latency increased at 14 days after exposure	WBA SAR 1.5, 3, 6 or 9 W/kg estimated from Wang et al. [16,20]	Li et al. [21]
MWM, OFA, EPM, tail suspension, forced swim CD-1 mouse 5 weeks old	9.417 GHz 200 V/m (SAR 2 W/kg) 12 h/day from gestational day 3.5 to 18 Free	Impaired learning and memory only in male mice. Increased anxiety and decreased depression in males and females	Basis of SAR calculation not given	Zhang et al. [22]
MWM Fischer 344 rat (*n* = 6) 180 days	900, 1800 or 2450 MHz 2 h/day, 5 days/week for 90 days WBA SAR 0.59, 0.58, 0.67 mW/kg Restrained	All exposures impaired performance in probe trial, with increased time to target quadrant and decreased time in quadrant	Also increased HSP70 levels and increased DNA strand breaks Identical results published by Deshmukh et al. [23] following exposure for 90 days	Deshmukh et al. [24]
MWM SD rat (*n* = 9) 220–250 g	900 MHz CW 3 h/day for 14 or 28 days Average SAR in the head 2 W/kg, WBA SAR 0.016 W/kg. Restrained	No effects on learning, exposure for 28 days significantly impaired memory	Ultrastructural changes and increased serum albumin leakage. Effects attributed to changes in mkp-1/ERK pathway	Tang et al. [25]
MWM Swiss mouse (*n* = 20) 12 weeks	2450 MHz CW 2 h/day for 15, 30 for 60 days WBA SAR 0.0146 W/kg Restrained	Significantly increased escape latency during acquisition, impaired memory in probe trial. Deficits increased with increasing exposure time	No effect on rectal temperature. Exposure time-dependent changes in neuronal morphology, apoptosis, oxidative state	Shahin et al. [26]
MWM Wistar rat (*n* = 25) 200 g	1500 MHz, 2856 MHz or both sequentially 6 min/frequency WBA SAR 1.8/1.7 W/kg or 3.7/3.3 W/kg Free	Escape latency increased only at higher SAR (both frequencies). No increased effect with sequential exposure	Skin temperature increased by <1 °C (*n* = 4). Changes in EEG and hippocampal morphology at higher SAR	Tan et al. [27]
MWM Wistar rat (*n* = 15) 8 weeks	2586 MHz, pulsed 6 min/day, 5 days/week for 6 weeks Brain SAR 1.7, 3.5 or 7 W/kg Restrained	Effects only at 7 W/kg: escape latency increased; impaired memory in probe trial.	No change in measured body temperature (*n* = 10). Changes in EEG, NMDAR, hippocampal structure	Wang et al. [28]
MWM Swiss mouse (*n* = 6) 14 days	1000 MHz 2 h/day for 15 days WBA SAR 0.179 W/kg Lightly restrained	Deficits in learning and memory when tested at 6 weeks of age	Biochemical and histological changes	Sharma et al. [29]
Radial arm maze, 4/8 task Swiss mouse (*n* = 20) 12 weeks	2450 MHz CW 2 h/day for 15, 30 or 60 days WBA SAR 14.6 mW/kg	Exposure-time dependent increase in errors in working and reference memory	No increase in rectal temperature	Shahin et al. [30]

Abbreviations: ANOVA: analysis of variance; CW: continuous wave; EPM: elevated plus maze; FTDT: finite-difference time-domain; GSM: Global System for Mobile communication; i.p.: intraperitoneal; MWM: Morris water maze; NMDAR: N-methyl-D-aspartate receptor; OFA: open field arena; PD: power density; SAR: specific energy absorption rate; SD: Sprague Dawley; WBA: whole-body average. “No effects” means no statistically significant effects. “Restrained” means that the animals were held immobile during exposure, and “free” means that the animals were free to move during exposure. The age and/or weight of the animals is given at the start of exposure.

**Table 2 ijerph-16-01607-t002:** Behavioural studies with RF fields reporting significant improvements of place learning and spatial memory.

Model	Exposure	Response	Comment	Reference
MWM, OFA, EPM Wistar rat (*n* = 18) 60.5–60.7 g (24 days)	900 MHz, GSM 2 h/day, 5 days/week for 5 weeks WBA SAR 0.3, 3 W/kg Free	Improved learning of escape platform with both SARs; improved memory of platform location with 3 W/kg	No effect on activity, anxiety, blood brain barrier	Kumlin et al. [31]
Y maze APPsw mouse, Tg or NT (*n* = 4–10) 21–26 months	918 MHz GSM E field 17–35 V/m 2 × 2 h/day, 1 month Free	Alternations in Y maze increased 26% when Tg and NT mice combined	SAR provided, but incorrectly calculated from external E field	Mori and Arendash [32]
MWM, radial arm maze, Y maze, circular platform APPsw mouse, Tg or NT (*n* = 5–9) 21–26 months	918 MHz GSM E field 17–35 V/m 2 × 2 h/day, up to 2 months Free	No field-dependent effects except alternations in Y maze increased when Tg and NT mice were combined (*p* < 0.05)	Exposure continued during testing period. No effects on agility, activity or exploration.SAR provided, but incorrectly calculated from external E field	Arendash et al. [33]
Y maze, OFA, passive avoidance tests 5xFAD mouse and WT (*n* = 7–11) 1.5 months	1950 MHz W-CDMA 2 h/day, 5 day/week for 8 months WBA SAR 5 W/kg Free	Decreased alternation in Y maze, increased time in centre of OFA, impairments in passive avoidance, all rescued by exposure; no effects on WT	Maximum increase of body temperature of 0.5 °C Aβ plagues and other pathology reduced by exposure	Jeong et al. [34]

See Table 1 for abbreviations, plus: NT: non-transgenic; Tg: transgenic; WT: wild type; W-CDMA: Wideband Code Division Multiple Access.

**Table 3 ijerph-16-01607-t003:** Behavioural studies with RF fields reporting an absence of significant effects on place learning and spatial memory.

Model	Exposure	Response	Comment	Reference
MWM, hippocampus morphology of offspring after exposure SD rat (*n* = 6) Day 3–18 of pregnancy (maze) or day 3–18 plus 1–10 postnatal day (morphology)	0.1–1 GHz ultra-wideband pulses 2 min/day, 16 days prenatally, 10 days postnatally WBA SAR 45 mW/kg Restrained	No effects on maze task (males only, on day 50), increased medial-to-lateral length of the hippocampus (day 21). Overall, no effects on 39 out of 42 endpoints	Clear responses to positive control (lead acetate). SAR calculated from power spectrum	Cobb et al. [35]
8-arm radial maze C57BL/6J mouse (*n* = 5) 12 weeks	900 MHz pulsed at 217 Hz 45 min/day, 10 days WBA SAR 0.05 W/kg Tested immediately or 15 or 30 min after exposure Restrained	No effects on performance	Animals tested immediately took longer to complete task both after RF and sham exposure. SAR calculated according to Johnson et al. [36]	Sienkiewicz et al. [37]
8-arm radial maze. spatial task in OFA SD rat (*n* = 8) 150 g	900 MHz pulsed at 217 Hz 45 min/day, 10 days (radial maze) or 14 days (spatial task) Brain average SAR 1 or 3.5 W/kg Restrained	No effects	Head-only exposure. SAR calculated from temperature measurements and using FDTD methods	Dubreuil et al. [38]
Two versions of 8-arm radial maze SD rat (*n* = 12 or 9) 120 g	900 MHz pulsed at 217 Hz 45 or 60 min/day, 4, 12 or 16 days Brain SAR 1 or 3.5 W/kg Restrained	No effects	Head-only exposure. SAR calculated from temperature measurements and using FDTD methods	Dubreuil et al. [39]
T-maze reversal learning SD rat (*n* = 15–28) 670 g	1439 MHz pulsed 6.7 ms pulses at 50 pps 45 or 60 min/day, 4 days or 60 min/day, 4 × 5 days Brain average SAR 7.5 or 25 W/kg Restrained	No effect on performance at lower SAR, decreased performance at higher SAR resulting in increased core temperature	Head-mainly exposure (animals positioned with head towards antenna). SAR calculated from temperature measurements and using FDTD methods	Yamaguchi et al. [40]
12-arm radial maze SD rat (*n* = 7 or 8) 250–300 g	2450 MHz pulsed; 2 μs pulses at 500 pps 45 min/day, 10 days WBA SAR 0.6 W/kg Restrained	No effects on performance and no effect of treatment with physostigmine, naltrexone or naloxone	Did not confirm Lai at al. [8]. SAR calculated using calorimetry and input/output difference	Cobb et al. [41]
12-arm radial maze SD rat (*n* = 12) 3 months, 270–320 g	2450 MHz pulsed; 2 μs pulses at 500 pps 45 min/day, 10 days WBA SAR 0.6 W/kg Restrained	No effects on performance in maze with access to distal spatial cues	Did not confirm Lai at al. [8]. SAR calculated using FDTD methods	Cassel et al. [42]
12-arm radial maze Sprague Dawley rat (*n* = 12) 3 months, 270–320 g	2450 MHz pulsed; 2 μs pulses at 500 pps 45 min/day, 10 days WBA SAR 2 W/kg Restrained	No effects on performance in maze with access to distal spatial cues	Did not confirm Lai et al. [8]. SAR calculated using FDTD methods.	Cosquer et al. [43]
12-arm radial maze SD rat (*n* = 12) 3 months, 270–320 g	2450 MHz pulsed; 2 μs pulses at 500 pps 45 min/day, 10 days WBA SAR 2 W/kg Restrained	No effects on performance in maze with reduced access to distal spatial cues	Did not confirm Lai et al. [8]. SAR calculated using FDTD methods	Cosquer et al. [44]
EPM SD rat (*n* = 12) 3 months, 270–320 g	2450 MHz pulsed; 2 μs pulses at 500 pps 45 min/day, 10 days WBA SAR 2 W/kg Restrained	No effect on anxiety with ambient light of 2.5 or 30 lux	SAR calculated using FDTD methods	Cosquer et al. [45]
8-arm radial maze over 10 days with further 8 days with 45 min inter trial delay after 4 correct responses SD rat (*n* = 6) 6 weeks	900 MHz GSM 45 min/day at average brain SAR 1.5 W/kg or 15 min/day at brain SAR 6 W/kg, 5 days/week, 8 or 24 weeks before testing Restrained	No effects	Head-only exposure. SAR calculated from temperature measurements and FDTD methods in Dubreuil et al. [39]	Ammari et al. [46]
MWM SD rat (*n* = 6) 2 days	840 MHz 3 h/day, 12 days PD 60 µW/m^2^ Free moving	No effects	Increased freezing behaviour in males (mood disturbance)	Daniels et al. [47]
MWM, OFA CR1:CD(SD) rat (*n* = 42–28) Adults: 10 weeks + 5 days acclimatization Offspring: 4 days	2140 MHz, W-CDMA 20 h/day, from day 7 of gestation to delivery and day 4–21 after birth Dams: WBA SAR 0.066–0.093, 0.028–0.04 W/kg; Foetus/progeny: WBA SAR 0.068–0.146, 0.029–0.067 W/kg Free	No effects on offspring	SAR calculated using FDTD methods	Takahashi et al. [48]
MWM, 8-arm radial maze (4/8 version), OFA Wistar rat (*n* = 20) 14 days	900 MHz GSM Average brain SAR 0.7, 2.5 or 10 W/kg 2 h/day, 5 days/week, 18 months Restrained	No effects observed as juveniles, adults or pre-senile (more specific ages not given)	Head-only exposure. No probe trial. SAR calculated using FDTD methods as in Spathman et al. [49]	Klose et al. [50]
Orientation response to magnetic north European robin (*n* = 18–42) Age not specified	Background fields 0.01–5 MHz, 1008 nT Screened condition 0.01–5 MHz, 2.56 nT Applied fields 0.01–5 MHz, 278 nT; 0.02–0.45 MHz, 133 nT; 0.6–3 MHz, 34 nT Free	No response with background fields, or with applied fields. Response with screen	Exposures given as accumulated time-dependent magnetic field intensity summed over relevant frequency range	Engels et al. [51]
MWM, OFA SD rat (*n* = 16) 9 weeks	2.14 GHz W-CDMA WBA SAR <0.24, <0.08 W/kg 20 h/day gestational day 7 to postnatal day 21 (weaning) then 24 h/day for 3 weeks Free	No consistent effects	SAR variable due to growth and movement No effects on development or anxiety	Shirai et al. [52]
Radial arm maze, OFA, EPM, fear conditioning Wistar rat (*n* = 6–8) 4–6 and 22–24 months	900 MHz 2 h/day, 5 days/week for 4 weeks Brain SAR 6 W/kg Restrained	Age-related differences. No field-related effects except for decrease in anxiety	No field-related changes in IL-1β, IL-6 or GFAP levels	Bouji et al. [53]
MWM, Y maze, OFA, object recognition Mice: 5xFAD (*n* = 8) 1.5 months old	1950 MHz 2 h/day, 5 days/week, for 3 months WBA SAR 5 W/kg Free	No effects	Measured rectal temperatures changed from −1.9 °C to +0.5 °C	Son et al. [54]
MWM, OFA, EPM C57/BL mouse (*n* = 10, 15) 4 weeks	1800 MHz 6 h/day for 28 days WBA SAR 2.7 W/kg, Brain SAR 2.2 W/kg Free	No effects except decreased behavioural anxiety	No increase in skin temperature Increased levels of GABA and Asp in cortex and hippocampus.	Zhang et al. [55]
MWM, OFA SD rat (*n* = 8) Gestational day 7	Multiple frequency signal, 880 to 5180 MHz 20 h/day from gestational day 7 to 6 weeks of age WBA SAR 0.08, 0.4 W/kg Free	No consistent effects	No consistent teratological or developmental effects	Shirai et al. [56]

See Table 1 for abbreviations plus: Asp: aspartic acid; GABA: γ-aminobutyric acid; GFAP: Glial fibrillary acidic protein.

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
