# Peer review of "Can Low-Level Exposure to Radiofrequency Fields Effect Cognitive Behaviour in Laboratory Animals? A Systematic Review of the Literature Related to Spatial Learning and Place Memory"

_ijerph, 2019, doi:10.3390/ijerph16091607_

Round 1
Reviewer 1 Report
The paper is a very well written systematic review. The authors should only pay attention to the fact that more information should be added in the reasons for non inclusion of the studies presented in the appendix according to the inclusion criteria presented in lines 80-87 of the paper.
Author Response
Point 1: The authors should only pay attention to the fact that more information should be added in the reasons for non inclusion of the studies presented in the appendix according to the inclusion criteria presented in lines 80-87 of the paper.
Response 1: We have added text in the appendix (original line 612) summarising the three criteria we utilised for defining non-inclusion. But regarding additional descriptions in the table, we strongly feel that the existing descriptions are self-explanatory and provide sufficient information to understand our reasons for excluding a study, and additional text is not really necessary. For example, if a study could not be included because of the absence of appropriate dosimetry, the reason for non-inclusion would simply be the lack of dosimetry.
Reviewer 2 Report
This is an interesting study about the effect of electromagnetic exposure in animals by means of a systematic review.
Some comments below that could improve the quality of the paper:
- The search criteria focuses on laboratory animals, this should perhaps be stated on the title of the paper, as behaviour of animals extends to any animal, and there are more studies not included that analyse the behaviours for example of cows when close to base-station antennas.
- Although PubMed focuses on biomedical entries, why the authors left away other scholarly databases such as Scopus, Web of Science, Proquest, ScienceDirect, Google Scholar,...? Some returns would have been missed by PubMed. I encourage the authors to reconsider other databases that may increase the 62 findings.
Author Response
Point 1: The search criteria focuses on laboratory animals, this should perhaps be stated on the title of the paper, as behaviour of animals extends to any animal, and there are more studies not included that analyse the behaviours for example of cows when close to base-station antennas.
Response 1: The first line of the abstract indicates that the review is concerned with laboratory studies, but for completeness we have added "laboratory" to the title as suggested.
Point 2: Although PubMed focuses on biomedical entries, why the authors left away other scholarly databases such as Scopus, Web of Science, Proquest, ScienceDirect, Google Scholar,...? Some returns would have been missed by PubMed. I encourage the authors to reconsider other databases that may increase the 62 findings.
Response 2: We also used EMF Portal for our search, which covers many additional sources and databases, and together with PubMed provide an comprehensive coverage of the relevant literature. More informally, we compared our results to previous reviews and commentaries in this area and we feel we have not missed salient, well performed studies. We have stated the databases and methods we used very clearly and openly in the text,
Reviewer 3 Report
The authors publish a detailed review of literature surrounding the effects of RF fields on laboratory animal behavior. The authors have compiled a significant amount of data, and their supplied narrative is thorough.
My only critique is a minor one and is more related to a missed opportunity, rather than a critique of the information they have presented. I believe the table they have compiled is full of information, but at the same time not very informational. The authors could take the opportunity to present their meta-analyses using various graphical illustrations. This would greatly enhance the readability and impact of the paper.
For one, I would be very interested to see a plot of the data describing the studies with regards to Frequency (x-axis) vs Power or SAR (y-axis), where each of the points is a unique paper experimental condition. The authors mention that a variety of frequencies were tested, but there is not an easy way to digest or get a quantitative feel for how varied it is. The authors could go even further and use different marker styles to denote some other feature of the papers included (e.g., conclusion of paper = positive, negative, or neutral). The latter could tremendously increase the ability of the reader to interpret whether the experimental conditions utilized for each study may have an impact on the conclusions.
Second, I would suggest a graphical summary of the animal models included in the review (Moris Maze, Radial arm, etc). It seems that a substantial conclusion of the review is that the varied findings in the literature are likely to stem from minutia and logistical challenges of conducting behavioral experiments. Having personally conducted rodent behavior, I am well aware at how the strangest environmental factor can influence a study (e.g., the shampoo the experimenter is using on the day of the test!).
I believe a narrative or infographic summary of the animal models could help support the contextual understanding of this paper. For example, for each of the animal models tested, briefly describe the objective, the test outcomes, and the potential limitations of each. This infographic could significantly enhance the readers ability to understand the context of the behavioral methods employed to study the RF effects.
I believe the manuscript is publishable as-is and already contains most of the requested information. My suggestions are for further improvement purposes only.
Author Response
Point 1: My only critique is a minor one and is more related to a missed opportunity, rather than a critique of the information they have presented. I believe the table they have compiled is full of information, but at the same time not very informational. The authors could take the opportunity to present their meta-analyses using various graphical illustrations. This would greatly enhance the readability and impact of the paper.
For one, I would be very interested to see a plot of the data describing the studies with regards to Frequency (x-axis) vs Power or SAR (y-axis), where each of the points is a unique paper experimental condition. The authors mention that a variety of frequencies were tested, but there is not an easy way to digest or get a quantitative feel for how varied it is. The authors could go even further and use different marker styles to denote some other feature of the papers included (e.g., conclusion of paper = positive, negative, or neutral). The latter could tremendously increase the ability of the reader to interpret whether the experimental conditions utilized for each study may have an impact on the conclusions.
Response 1: We have addressed the point by rearranging the narrative into separate sections defined by task and species. Plus we have added a figure as suggested to show the relationship of frequencies and SARs used with studies denoted by outcome. In this way we believe have made the data more informative and assessible to the reader
Point 2: Second, I would suggest a graphical summary of the animal models included in the review (Moris Maze, Radial arm, etc). It seems that a substantial conclusion of the review is that the varied findings in the literature are likely to stem from minutia and logistical challenges of conducting behavioral experiments. Having personally conducted rodent behavior, I am well aware at how the strangest environmental factor can influence a study (e.g., the shampoo the experimenter is using on the day of the test!).
Response 2: We have not attempted the graphical summary, as the narrative has been substantially rearranged by task (and species) and short summaries have also been included now. While some of the details of how the tasks were performed task are included in the papers, and many minutiae are unknown, making explicit comparisons impossible.
Point 3: I believe a narrative or infographic summary of the animal models could help support the contextual understanding of this paper. For example, for each of the animal models tested, briefly describe the objective, the test outcomes, and the potential limitations of each. This infographic could significantly enhance the readers ability to understand the context of the behavioral methods employed to study the RF effects.
Response 3: We have added a narrative summary to each of the subsections of the narrative to increase understanding. However, given the large amount of information that is available on the internet and in published texts,regarding the best practices for behaviour, and the limitations of the tasks used, we strongly feel that it is not necessary to include such a primer on behavioural techniques here. We have already briefly mentioned main features of the major tasks and their objectives.